# Flow Resistance in Open Channel Due to Vegetation at Reach Scale: A Review

**Antonino D'Ippolito \*, Francesco Calomino, Giancarlo Alfonsi and Agostino Lauria**

Dipartimento di Ingegneria Civile, Università della Calabria, 87036 Arcavacata, Italy; francesco.calomino@unical.it (F.C.); giancarlo.alfonsi@unical.it (G.A.); agostino.lauria@unical.it (A.L.)
\* Correspondence: antonino.dippolito@unical.it Tel.: +39-0984-496550

**Abstract:** Vegetation on the banks and flooding areas of watercourses significantly affects energy losses. To take the latter into account, computational models make use of resistance coefficients based on the evaluation of bed and walls roughness besides the resistance to flow offered by vegetation. This paper, after summarizing the classical approaches based on descriptions and pictures, considers the recent advancements related to the analytical methods relative both to rigid and flexible vegetation. In particular, emergent rigid vegetation is first analyzed by focusing on the methods for determining the drag coefficient, then submerged rigid vegetation is analyzed, highlighting briefly the principles on which the different models are based and recalling the comparisons made in the literature. Then, the models used in the case of both emergent and submerged rigid vegetation are highlighted. As to flexible vegetation, the paper reminds first the flow conditions that cause the vegetation to lay on the channel bed, and then the classical resistance laws that were developed for the design of irrigation canals. The most recent developments in the case of submerged and emergent flexible vegetation are then presented. Since turbulence studies should be considered as the basis of flow resistance, even though the path toward practical use is still long, the new developments in the field of 3D numerical methods are briefly reviewed, presently used to assess the characteristics of turbulence and the transport of sediments and pollutants. The use of remote sensing to map riparian vegetation and estimating biomechanical parameters is briefly analyzed. Finally, some applications are presented, aimed at highlighting, in real cases, the influence exerted by vegetation on water depth and maintenance interventions.

**Keywords:** river hydraulics; vegetation; flow resistance; turbulence; numerical methods.

## 1. Introduction

As it is well known, vegetation is an important issue on the viewpoint of catchment hydrology [1–3], since rain drops interception, evapotranspiration, and infiltration are elements to consider in surface and sub-surface water balance.

Moreover, riparian vegetation plays a key role both on the ecologic and habitat viewpoints, as well as a source of biodiversity. Indeed, vegetation creates micro-environments that can host birds, small mammals, and insects helpful to agricultural purposes, prevents fertilizers and pollutants from getting to the watercourses [4] and, because of the effect on landscape, has a significant recreational function.

On a more strictly technical viewpoint, riparian vegetation interacts with water flow, with effects both on the bank stability and on the river hydraulics. Vegetation acts on bank stability since it mechanically strengthens the soil because of the presence of roots [5–8]; moreover, it reduces the soil water content because of evapotranspiration, with the consequence of reducing interstitial pressures [9].

As for river hydraulics, vegetation clutters up part of the river cross-section [7,10,11], increases the roughness, and reduces the velocity; all this results in increased

water levels and reduced water conveyance. Moreover, while the smaller average velocity on one hand reduces the erosion of the riverbed and banks, on the other one, it increases the sediment deposition, what makes the water cross-sections smaller and raises flooding risk. On the scale of the hydrographic network, the general velocity reduction influences the travel time of water particles, making the peak flow control easier [12,13].

Therefore, one cannot know the general effect of vegetation in advance, but every case should be considered singularly, using proper procedures. Indeed, this effect depends on both the hydraulic and mechanical properties of the water cross-section, as of the present vegetation, that may be different according to species, phenological stage, age and, possibly, maintenance.

Numerous studies are presented in the literature on the experimental, theoretical, and computational points of view [4,14–19]. The major difficulty lies in the impossibility of studying the problem under common conditions or to draw conclusions of a general value from the experiments or from the case studies. The recent progresses in the numerical solution of the flow equations make it possible to treat single problems, but it remains difficult to adapt the codes to different conditions.

Usually, in the literature, the vegetation is considered as rigid or flexible, and according to the water level, as emergent or submerged. Flexible vegetation refers to grass, reeds, and shrubs, or, when speaking about trees, to the branch and leaf system. Combinations of the above categories, really found in natural streams and channels, are still difficult to treat.

One should note that, commonly, the river cross-sections present variable roughness along the wetted perimeter, and a typical example is given by vegetated cross-sections; moreover, one river station can be formed by more sub cross-sections, differently shaped. When using one-dimensional models of water flow, one needs a roughness coefficient representative enough that it can be computed as a weighted average of local roughness coefficients. To this purpose, several equations are present in [20] on the basis of the assumptions made on how a particular variable (discharge, velocity, contour shear stress) in the subsections is related to that in the total section.

Although the subject of this review is the flow resistance due to vegetation, however, it should be noted that research on the interaction between vegetation and flow is currently focusing on a more correct assessment of the action exerted by the shear stress on the bed and the banks [21,22], on velocity distribution [23,24], on sediment transport [25–31], on finite-sized vegetation patches [32–39], on the interaction between jets and vegetation [40,41], on processes of transport and dispersion [42–44], on evolution of patches of vegetation [45], and on one-line emergent vegetation [46].

Correctly evaluating the resistance to flow is then a major aspect of not only the river safety, regulation, and maintenance, but also of flood model calibration and validation. In the following, we will present the methods found in the literature, allowing estimation of flow resistance coefficients to input into models for flood simulation, based on different types of vegetation in the river banks and floodplains. We will then analyze briefly the three-dimensional modeling of free surface flow in the presence of vegetation and the use of remote sensing for mapping riparian vegetation and estimating biomechanical parameters. Finally, the issue of vegetation management will be dealt with, presenting a number of points of view when cutting or pruning of plants is required for safety or maintenance reasons.

## 2. Flow Resistance Equations

According to Chow [47], the resistance to flow in artificial channels and watercourses is influenced by several factors, i.e., size and shape of the grains of the material forming the wetted perimeter, vegetation, channel irregularity, channel alignment, silting and scouring, obstruction, size and shape of channel, stage and discharge, seasonal change, suspended material, and bed load. More classifications exist, among which the best known are those by Rouse [48] and Yen [20].

As is well known, the resistance to flow can be expressed by the Darcy-Weisbach $f$ friction factor, the Chézy's $C$, or the Gauckler-Strickler $k$ velocity coefficients and the Manning $n$ roughness coefficient; the relation among these coefficients is the following:

$$\sqrt{\frac{8}{f}} = \frac{R^{1/6}}{n\sqrt{g}} = \frac{kR^{1/6}}{\sqrt{g}} = \frac{C}{\sqrt{g}} = \frac{V}{\sqrt{gRJ}} \qquad (1)$$

where $V$ is the mean flow velocity, $R$ the hydraulic radius, $J$ the energy line slope, and $g$ the gravity acceleration.

Cowan [49], considering the case of fixed bed and ignoring the effects of the suspended solid flux on the turbulence, believes that the resistance to flow in natural water-courses depends on one hand on the shape, the dimension, and arrangement of the elements that form the roughness; and on the other hand, it depends on further dissipative effects, caused by macro-vortices produced by the flow separation in the abrupt changes of direction, cross-section shape, and vegetation; and that the overall Manning coefficient can be expressed as the sum of the relevant values.

The contribution of vegetation to the roughness coefficient can be evaluated by means of descriptive approaches, photographic comparison approaches, or by analytical methods. Even though the descriptive and photographic approaches should be considered empirical in nature, nevertheless they add to the knowledge of the river conditions and in many cases are a good way to assess a value of a friction factor or roughness/velocity coefficient.

## 3. Descriptive and Photographic Comparison Approaches

In these methods, one roughness coefficient is chosen on the basis of the class to which the river reach belongs. Among the descriptive methods, the best known is Chow's [47]. The author gives, for every class of channels, the minimum, average, and maximum values of Manning $n$ coefficient, warning that when the channel is artificial, the average values should be used in case of good maintenance only. In Tables 5–6 of Chow's book ([47], p. 110–113), one can observe that the Manning coefficient is 0.018 sm$^{-1/3}$ in case of the excavated channel, straight, clean, uniform cross-section with no vegetation, and 0.035 sm$^{-1/3}$ in case of dense weeds. In natural streams, its values are 0.030 sm$^{-1/3}$ when the cross-section is clean, and 0.045 sm$^{-1/3}$ in case of weeds. We should underline that for banks with many trees, the normal values of the Manning coefficient vary in a rather large field, between 0.040 sm$^{-1/3}$ and 0.150 sm$^{-1/3}$.

The photographic comparison approach consists of evaluating the Manning coefficient of a given river reach on the basis of similarity with the pictures of other similar cases, for which the coefficient was estimated in ordinary or flood conditions. Chow [47] shows some pictures that can be considered as a first example, representing 24 cases concerning typical channels and one natural river, with a short description of the channel conditions and the corresponding value of Manning coefficients.

Some more manuals show color pictures carrying data and description of rivers in the USA and Australia [50–53], with features very different from each other and Manning coefficients ranging in a wide field. In addition to a qualitative description of the channel, there is some information about cross-sections and hydraulic characteristics. Almost all the watercourses exhibit riparian vegetation over the banks, but estimates of the Manning coefficients very often refer to the main channel only; indeed, the values pertaining to flooding cases are limited.

Arcement and Schneider [54] show the pictures of 15 areas subject to flooding and densely vegetated, for which roughness coefficients are evaluated. This is one of the few works trying to define the contribution of vegetation to the total flow resistance. Starting from the Cowan [49] approach and taking into account the method of vegetation density proposed by Petryk and Bosmajian [55], the authors make use of density measurements and of an effective drag coefficient, to compute the total roughness and the different

contribution to it. The contribution due to vegetation ranges between percentages of the total roughness from 64% to 81%, that is very significant values.

Some researchers think that the above methods are probably more reliable than the analytical, since, in field observations, the heterogeneity is accounted for. Nevertheless, pictures of vegetated channels are more or less limited, and, consequently, choosing a reference channel is often difficult, not to say impossible.

## 4. Analytical Methods

In the analytical methods, plants are generally described by a characteristic diameter, $D$, and height, $h_v$, or by a characteristic area of plants, $A_c$. These methods are mainly suitable in laboratory experiments, where reference is made to both natural and artificial vegetation, the latter represented by cylinders of different materials or strips of plastic. Much more limited, in this approach, are the field experiments.

In laboratory experiments, the elements representing vegetation are often distributed with parallel (called linear by some authors) or staggered patterns; there are cases where the distribution is random, as vegetation usually is in nature (Figure 1).

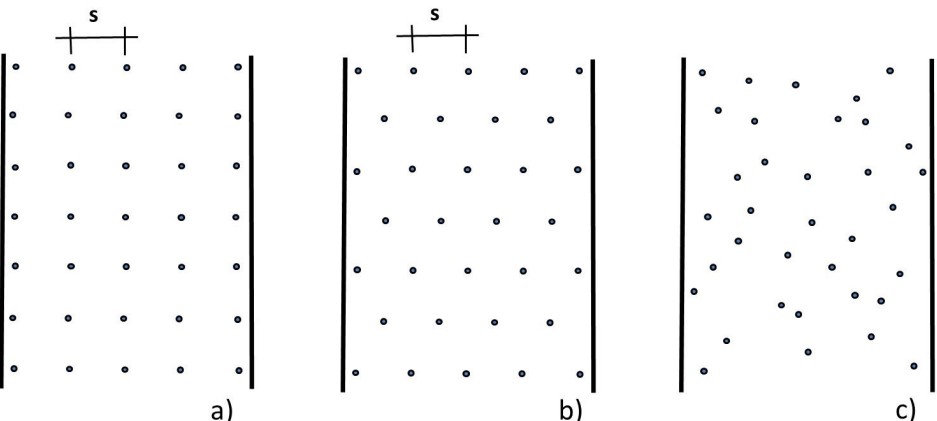

**Figure 1.** Plan view of (**a**) parallel, (**b**) staggered, and (**c**) random patterns.

In the case of rigid vegetation represented by cylinders, the density of vegetation $\lambda$ is often expressed as

$$\lambda = \frac{m\pi D^2}{4} \tag{2}$$

where $m$ is the number of cylinders per unit bed area; it is also used as a density measure of the projected plant area per unit volume, $a$, with

$$a = mD \tag{3}$$

as well as the ratio of stem diameter to spacing between the stems $D/s$.

### 4.1 Rigid Vegetation

When the vegetation is made up by trees, in an analogy of the resistance to flow due to immersed bodies, the roughness coefficient is expressed as a function of the drag force exerted by the flow on the body, depending then on the number of trees, their arrangement, the diameters of their trunks and, where appropriate, the branch system. Usually, in the laboratory tests, the rigid vegetation is simulated by cylinders, like in Figure 2, which is practically correct when the flow does not touch the leafage.

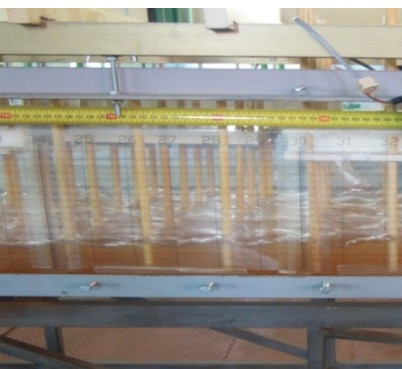

**Figure 2.** Laboratory flume with arrays of cylinders representing rigid vegetation, University of Calabria.

### 4.1.1. Emergent Rigid Vegetation

In case of one isolated vertical cylinder whose axis is orthogonal to the flow direction, the resistance to flow is expressed by the drag force $F_D$, computed as

$$F_D = \frac{\rho C_D h D V^2}{2} \tag{4}$$

where $\rho$ is the water density, $h$ is the depth of the immersed part of the cylinder, $V$ is the approach velocity, and $C_D$ is a drag coefficient. $C_D$ is a function of a stem Reynolds number computed by means of the approach velocity $V$ and the cylinder diameter $D$, $Re_D = \frac{VD}{\nu}$, $\nu$ being the water kinematic viscosity.

When the cylinder is a part of a group of elements (see Figure 3), one cannot ignore that the longitudinal and transversal interference make considerably more difficult the study of the resistance to flow [56–58]. In Figure 3, $x$ is the streamwise coordinate, $z$ is the vertical coordinate above the river bed, $u_z$ is the local time-averaged velocity, $u$ is the mean velocity along the vertical, $h$ is the water depth, and $h_v$ the vegetation height.

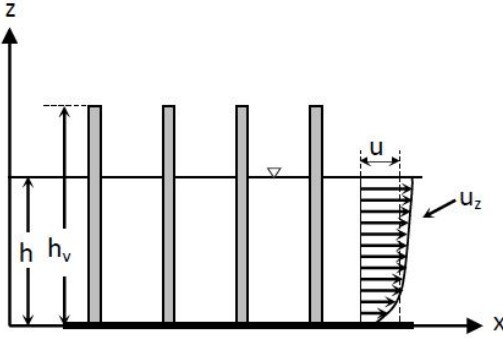

**Figure 3.** Side view of emergent vegetation and velocity profile.

Liu et al. [59], based on an experimental survey, described in detail the flow characteristics through rigid vegetation both in emergent and submerged conditions; they considered a staggered and a linear arrangement and for the latter referred to different densities; they also considered bottoms with different roughness. Liu et al. [59] measured the horizontal component of the velocity on several verticals between two dowels and also in the free field.

With reference to the emergent vegetation in the verticals immediately downstream of the dowels, the lower velocities are obtained; the velocities are progressively increasing, proceeding downstream and approaching the next dowel. In the free field there are

higher velocities. Along each vertical, the velocities are practically constant for most of the height and gradually increase as they approach the free surface. In the case of verticals placed between two dowels, the velocity distribution near the bed has a spike which is particularly pronounced in the vertical immediately downstream of the dowel and which attenuates in the flow direction.

The velocity spike is probably caused by the horseshoe vortex that forms at the base of the dowel, attracting the liquid of the surrounding region, which is faster, towards the base of the dowel. The fluid masses near the bottom, characterized by high speed values, mix with the higher, characterized by low speeds, creating vortices that rotate counterclockwise. These vortices, moving downstream, bring the fluid masses upwards, and this determines, as the abscissa increases, the reduction of the velocity spike. The staggered distributions determine a greater resistance than the linear [56,59].

The tests carried out with a rough bed have shown a significant change in the velocity distribution only as regards the profile immediately downstream of the dowel where there have been reductions in velocity from 30 to 130%, and also an increase of about 20% for the velocity spike near the channel bed. The measurements of the vertical component of the velocity have shown, in the vertical immediately downstream to the dowel and in the vicinity of the bed, a relatively large value which is attenuated by proceeding in the flow direction. In the vertical downstream to the dowel, Liu et al. [59] found weak downward velocity components.

As to the turbulence intensity relative to the streamwise component of the velocity, Liu et al. [59] found almost constant values along each vertical, however, its magnitude varied significantly with the location, presenting the highest values in the verticals immediately downstream to the dowels and the smaller ones in the free stream region. The turbulence generated by the dowels has a length scale much smaller than the shear generated turbulence and is quickly dissipated. In the free stream region, the turbulence intensity increases near the bed due to the shear.

To find the values of $C_D$ in case of sparse emergent arrays, both experimental tests and numerical simulations have been carried out, with the vegetation arrangements defined before as linear, or staggered, or random. Among the first studies, we will cite only Petryk [60], Li and Shen [56], and Petryk and Bosmajian [55].

Petryk and Bosmajian [55], to determine the Manning coefficient in a vegetated channel, implement the momentum equation for a reach, imposing that the vector sum of the weight of the control volume be equal to zero, projected on the bed direction, plus the contour resistance, plus that opposed by the tree trunk; they conclude with defining the overall Manning coefficient $n$ as a function of the value relative to the soil, $n_b$, and the one relative to vegetation drag coefficient $C_D$, by writing

$$n = n_b \sqrt{1 + \frac{C_D \sum A_i}{2gAL} \left(\frac{1}{n_b}\right)^2 R^{4/3}} \tag{5}$$

where $L$ is the reach length, $A$ the area of the water cross-section, $\sum A_i$ the area opposed by the vegetation to the flow. The authors consider the drag coefficient $C_D$ equal to 1.

Stone and Shen [61] developed a method for predicting the apparent channel velocity and the velocities in the stem layer in both emergent and submerged conditions. Four staggered arrangements of stems with varying diameter and density were used. The drag coefficient values compare well with that for a single cylinder, and the authors suggest an average value of 1.05-applicable with the average velocity in the constricted cross-section.

Baptist et al. [62], based on the balance of horizontal momentum in uniform steady flow condition, calculated the Chézy coefficient in the presence of emergent vegetation, $C_k$, by writing

$$C_k = \sqrt{\frac{1}{(1/C_b^2) + (C_D mDh/2g)}} \tag{6}$$

where $C_b$ is the Chézy coefficient of the bed. Baptist et al. [62] consider $C_D = 1$.

In Table 1 are reported different expressions for the drag coefficient. In the case of Ishikawa et al. [63], Kothiary et al. [64], and D'Ippolito et al. [65] they are based on direct measurements of the drag force. In Kothyari et al. [64], the cylinders are distributed on a grid with a staggered pattern forming angles of 30° with the flow direction, and the action is on one cylinder, while in the case of Ishikawa et al. [63], the angle with the flow direction was 45° and the action was calculated as mean on seven or thirteen cylinders, while instead in D'Ippolito et al. [65], the cylinders are in a linear arrangement and the action was calculated as mean on two to twenty five cylinders.

**Table 1.** Equations for estimating the drag coefficient $C_D$ in case of arrays of emergent cylinders.

| Authors | Relationship |
|---|---|
| Ishikawa et al. [63] | $C_D = 1.71\lambda^{0.11}$ when $i = 1/50$;<br>$C_D = 2.45\lambda^{0.20}$ when $i = 1/20$;<br>$C_D = 3.89\lambda^{0.31}$ when $i = 1/10$; |
| Tanino and Nepf [66] | $C_D = 2\left(\dfrac{\alpha_{0E}}{Re_{D*}} + \alpha_{1E}\right)$ |
| Kothyari et al. [64] | $C_D = 1.53[1 + 0.45\,ln(1 + 100\lambda)]Re_{D*}^{-3/50}$ |
| Cheng and Nguyen [67] | $C_D = \dfrac{50}{Re_v^{0.43}} + 0.7\left[1 - exp\left(-\dfrac{Re_v}{15,000}\right)\right]$  for $Re_v = 52$ - $5.6{*}10^5$<br>with $Re_v = \dfrac{V_v r_v}{\nu}$ and $r_v = \dfrac{\pi}{4}\dfrac{(1-\lambda)}{\lambda}D$ |
| Cheng and Nguyen [67] | $C_D = \dfrac{130}{r_{v*}^{0.85}} + 0.8\left[1 - exp\left(-\dfrac{r_{v*}}{400}\right)\right]$  for $r_{v*} = 24 - 5,000$<br>with $r_{v*} = \left(\dfrac{g J}{\nu^2}\right)^{1/3} r_v$ |
| Wang et al. [68] | $C_D = \dfrac{50}{Re_v^{0.5}} + 4.5\dfrac{D}{h} - 0.303 ln\lambda - 0.9$ |
| Sonnenwald et al. [69] | $C_D = 2\left(\dfrac{6475D + 32}{Re_{D*}} + 17D + 3.2\lambda + 0.5\right)$ |
| D'Ippolito et al. [65] | $C_D = 0.211\,ln(100\lambda) + 0.784$<br>$0.003 < \lambda < 0.05$, $0.48\% < i < 2.02\%$, $1000 < Re_D < 10.000$ |

($i$ is the bed slope, $\alpha_{0E}$ and $\alpha_{1E}$ are the Ergun coefficients, $J$ is the energy slope, $V_v$ is the average pore velocity $V_v = V/(1-\lambda)$, $r_v$ is the vegetation-related hydraulic radius $r_v = \frac{\pi}{4}\frac{(1-\lambda)}{\lambda}D$, $r_{v*}$ is the dimensionless vegetation-related hydraulic radius $r_{v*} = \left(\frac{gJ}{\nu^2}\right)^{1/3} r_v$, $Re_v$ is the vegetation Reynolds number $Re_v = \frac{V_v r_v}{\nu}$, $Re_{D*}$ is the stem Reynolds number calculated with the average pore velocity $Re_{D*} = V_v D/\nu$).

In Ishikawa et al.'s tests [63], the drag coefficients based on the experimental results differ significantly for the same stem Reynolds number, although the dependence is unclear. The authors give three equations for the drag coefficient as a function of the vegetation density, depending on the flume slope.

Tanino and Nepf [66], on the basis of Ergun's [70] studies, proposed the equation reported in Table 1, where $\alpha_{0E}$ is relative to the contribution of viscous forces on the cylinder surface, and $\alpha_{1E}$ to the contribution of inertial forces deriving by the pressure drop downstream to the cylinders. The authors find that $\alpha_{1E}$ is linearly varying with the density, whereas $\alpha_{0E}$ is independent of the cylinder array characteristics for $0.15 \leq a \leq 0.35$. These results are confirmed by Tinoco and Cowen [71] for $Re_D > 1000$.

In the tests performed by Kothyari et al. [64], the density $\lambda$ is defined as Equation (2) and is ranging between 0.0022 and 0.0885. The mean flow velocity was estimated as the flowrate ($Q$) divided by the flume cross section and ($1-\lambda$), obtaining the so-called pore velocity $V_v$:

$$V_v = \frac{Q}{A(1-\lambda)},\tag{7}$$

In the case of subcritical flows, the authors obtained the equation reported in Table 1 in which $C_D$ remarkably increases with $\lambda$ and varies weakly with the stem Reynolds number calculated with the average pore velocity. The logarithmic increase of $C_D$ with $\lambda$

ensures that, when $\lambda$ is small, $C_D$ increases rapidly with $\lambda$, whereas, as $\lambda$ becomes large, $C_D$ tends to a constant value.

Cheng and Nguyen [67], when the wall and bottom effects are negligible, introduced the vegetation-related hydraulic radius $r_v = \frac{\pi}{4} \frac{(1-\lambda)}{\lambda} D$, which is a function of the vegetation density and diameter only. This vegetation-related hydraulic radius is used together with the pore velocity to define a new Reynolds number, the vegetation Reynolds number $Re_v = \frac{V_v r_v}{\nu}$. Using experimental data from several authors (random, staggered, only two cases linear), suitably unified, they showed that the drag coefficient decreases monotonically with the increase in the vegetation Reynolds number and propose the two equations reported in Table 1, respectively, function of $Re_v$ and $r_{v*}$ with $r_{v*}$ dimensionless vegetation-related hydraulic radius.

Wang et al. [68], in a study for incipient bed shear stress partition in mobile bed channels, investigated the vegetation drag coefficient. By ignoring the bed surface shear stress, an empirical formula was developed by data fitting in which the drag coefficient is a function of the Reynolds number, calculated with the vegetation-related hydraulic radius of Cheng and Nguyen [67], the ratio between vegetation diameter and flow depth (*D/h*), and the vegetation density $\lambda$. The proposed formula is reported in Table 1.

Sonnenwald et al. [69] based on the data of Ben Meftah and Mossa [23], Stoesser et al. [72], Tanino and Nepf [66], and Tinoco and Cowen [71] proposed the equation reported in Table 1, where the coefficient of the $D$ terms must have units m$^{-1}$ to retain non-dimensionality.

D'Ippolito et al. [65], on the basis of 70 tests with emergent stems in a linear arrangement, proposed an equation in which $C_D$ is a function of the density $\lambda$ only, valid in the field $\lambda$ from 0.003 to 0.05, bed slope $i$ from 0.48% to 2.02%, and $Re_D$ from 1000 to 10.000.

Figure 4 shows how the drag coefficient varies with density, and Reynolds numbers, for some of the formulas reported in Table 1. The values obtained by D'Ippolito et al. [65] are smaller with respect to those of other authors with the same $\lambda$, because of the different rod arrangements (square mesh against triangular mesh).

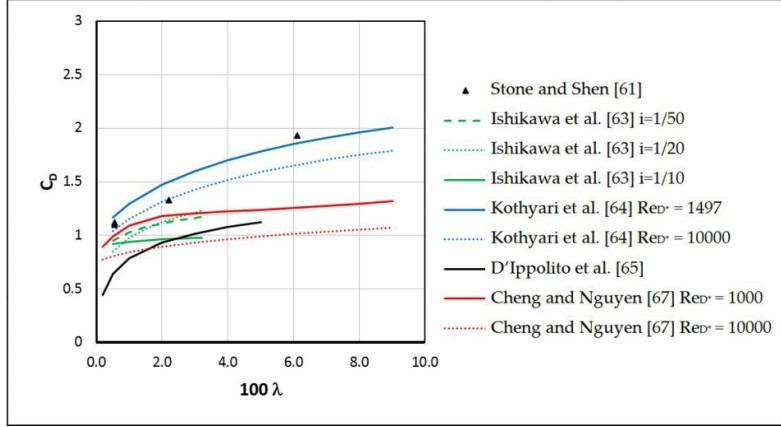

**Figure 4.** Emergent rigid vegetation—$C_D$ as a function of $\lambda$ and $Re_D$.

As it is easily seen, the $C_D$ values most frequently range between 0.5 and 2.0. One must notice that a comparison among the above equations is difficult, because in the equations of Kothyari et al. [64], $C_D$ has the form $C_D = C_D(\lambda, Re_{D*})$, while the equations of Cheng and Nguyen [67] have the forms $C_D = C_D(Re_v)$ and $C_D = C_D(r_{v*})$; the equation of Wang et al. [68] has the form $C_D = C_D(\lambda, D/h, Re_v)$; in the equation of Sonnenwald et al. [69], it is $C_D = C_D(\lambda, Re_{D*}, D)$; and in the equation of D'Ippolito et al. [65], $C_D = C_D(\lambda)$.

### 4.1.2. Submerged Rigid Vegetation

In the case of submerged vegetation, Baptist et al. [62] have identified in the velocity profile along a vertical four distinct areas, even though very often the profile is schematized as only two interacting zones (two-layer approach): The vegetation layer, containing the cylindrical elements representing the vegetation, and the surface layer, above them, up to the flow surface (Figure 5). In Figure 5, $u_s$ is the mean velocity along the vertical in the surface layer, and $u_v$ is the mean velocity along the vertical in the vegetated layer.

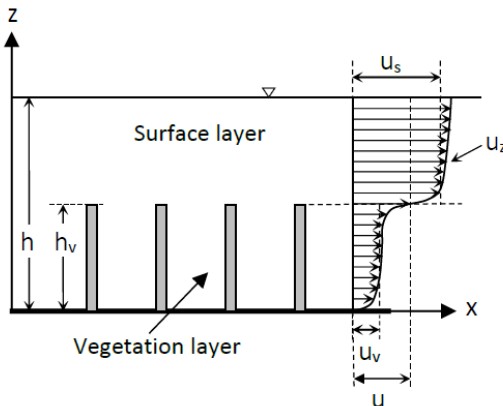

**Figure 5.** Side view of submerged vegetation and velocity profile.

The flow characteristics within a set of submerged cylinders, with the same arrangement and height of the cylinders, are similar to the case of emergent vegetation [59]. Above the cylinders, the liquid flows with a higher velocity and this, at the top of the cylinders, gives rise to an inflection point. The two coflowing streams, the upper one and the one between the cylinders, give rise to a Kelvin-Helmholtz instability, which causes the liquid to rotate clockwise, causing vortices that become larger in the downstream direction, forcing the inflection point deeper into the array. In the case of sparse vegetation, the vortex affects the entire vegetated layer, whereas in the case of dense vegetation, it affects only a layer limited to the top of the dowels [7]. The longitudinal turbulence intensities reach a maximum near the top of the dowel array. It has the largest values immediately behind the dowels and decreases in the flow direction. In the free stream region, the longitudinal turbulence intensity is lowest.

Rigid submerged vegetation has been the subject of a large number of investigations [7,61,62,73–77] and comparisons [78–83]. Some researchers provided the average velocity values in the two layers, while others derived the velocity distribution and the average values [7,73,74,76]. In the vegetation layer, the streamwise velocity is usually considered constant with the flow depth [62,76], while in the surface layer various expressions were adopted for the velocity distribution [81]: The logarithmic theory [61,73,76], the Kolmogorov theory of turbulence [75], the genetic programming [62], and the representative roughness height [77,84]. Usually, to determine the constants involved in the velocity formulas, the two distributions are required to assume the same values on the separation surface between the vegetated and the surface layer. The average velocity over the entire water depth is obtained as a combination of the velocity of the vegetated layer and that of the surface layer.

Klopstra et al. [73] derive the velocity profile in the vegetated layer starting from the turbulent shear stress, described by means of a Boussinesq-type equation, and adopt the logarithmic law for the surface layer. The different constants within the model are found on the basis of three conditions at the interface (continuity of shear stress, velocity and its vertical gradient) and a condition at the bed (negligible shear stress). The value of the Chézy coefficient, derived by Klopstra et al. [73], is a rather lengthy expression, and for it

we suggest referring to the original paper ([73], Equation (19)). The model has only one unknown parameter, the characteristic turbulence length scale, $\alpha_{KL}$, for which the following expression is proposed:

$$\alpha_{KL} = 0.0793 \, h_v \, ln\frac{h}{h_v} - 0.0090 \ \text{ for } \ \alpha_{KL} \geq 0.001 \tag{8}$$

For this parameter, van Velzen et al. [85] instead proposed the following relationship:

$$\alpha_{KL} = 0.0227 \, h_v^{\,0.7} \tag{9}$$

The equations for the determination of the Chézy coefficient in the case of submerged rigid vegetation, $C_r$, proposed by Baptist et al. [62] and Cheng [84] are shown in Table 2. This table also shows the Chézy coefficient obtained from the value of the average velocity proposed by Huthoff et al. [75] and the Manning coefficient proposed by Yang and Choi [76].

**Table 2.** Equations for estimating $C_r$ and $n$ in case of submerged rigid vegetation.

| Authors | Relationship |
|---|---|
| Baptist et al. [62] | $C_r = \sqrt{\dfrac{1}{(1/C_b^2)+(C_D mDh_v/2g)}} + \dfrac{\sqrt{g}}{\kappa} ln\left(\dfrac{h}{h_v}\right)$ |
| Huthoff et al. [75] | $C_r = \sqrt{\dfrac{2g}{C_D mDh}\left(\sqrt{\dfrac{h_v}{h}} + \dfrac{h-h_v}{h}\left(\dfrac{h-h_v}{1/\sqrt{m}}\right)^{\frac{2}{3}\left(1-\left(\frac{h}{h_v}\right)^{-5}\right)}\right)}$ |
| Yang and Choi [76] | $n = \left[\sqrt{\dfrac{2gh}{C_D ah_v}} + \dfrac{C_u\sqrt{g(h-h_v)}}{\kappa}\left(ln\dfrac{h}{h_v} - \dfrac{h-h_v}{h}\right)\right]^{-1} h^{2/3}$ |
| Cheng [84] | $C_r = \sqrt{\dfrac{\pi g}{2C_D}\dfrac{(1-\lambda)^3}{\lambda}\dfrac{D}{h_v}\left(\dfrac{h_v}{h}\right)^{3/2}} + 4.54\sqrt{g}\left(\dfrac{h-h_v}{D}\dfrac{1-\lambda}{\lambda}\right)^{1/16}\left(\dfrac{h-h_v}{h}\right)^{3/2}$ |

For submerged rigid vegetation, Baptist et al. [62] derived an equation by dimensionally aware genetic programming, and this equation has been improved manually.

Yang and Choi [76] proposed a two-layer model. The momentum balance was applied to each layer and an expression for the mean velocities was proposed. The velocity was assumed uniform in the vegetation layer and logarithmic in the upper layer. In the equation reported in Table 2, $C_u = 1$ for $a \leq 5 \, m^{-1}$ and $C_u = 2$ for $a > 5 \, m^{-1}$.

A representative roughness height characterized by its proportionality to both the stem diameter and vegetation concentration was proposed by Cheng [84] to estimate, with the same approach of Yang and Choi [76], the average flow velocity, and thus the resistance coefficients in vegetated channels. Cheng's [84] approach was developed for submerged rigid vegetation, but also gave good results in the case of flexible vegetation.

López and García [86] employed two numerical algorithms, the $k - \varepsilon$ and $k - \omega$ type, based on two closure equations, to model the mean flow and the turbulence structure in open-channel flows with submerged vegetation. The Manning coefficient, computed on the basis of experimental observations, shows an almost constant value close to the one corresponding to non-vegetated channels up to some threshold plant density and, once this limit is exceeded, a linear increase.

Defina and Bixio [74] extended the model by Klopstra et al. [73] and that by López and García [86] to consider both the plant geometry and drag coefficient variable with depth. A comparison between experimental data of real and artificial vegetation and the results of numerical simulations demonstrates that both models are able to reproduce the vertical profiles of velocity and shear stress within and above vegetation, whereas the turbulence characteristics are poorly predicted.

Li et al. [77] proposed the concept of the auxiliary bed and produced a dynamic two-layer model consisting of a basal layer and a suspension layer (Figure 6). The roughness of the suspension layer is defined as the product of the concentration of veg-

etation, $\lambda$, and the depth of penetration of the suspension layer in the vegetation, $\delta_e$. The authors obtained the following expression for the Manning coefficient:

$$n = \frac{h^{5/3}}{\sqrt{g}} \left[ \frac{1.96(h - h_v + \delta_e)^{5/3}}{(\lambda\delta_e)^{1/6}} + (1 - \lambda)(h_v - \delta_e)h_*^{1/2} \right]^{-1} \tag{10}$$

where $h_* = 2(1 - \lambda)/C_D a$, with $C_D$ calculated with the expression of Cheng and Nguyen [67]. For $\delta_e$, refer to Equation (7) of Li et al. [77].

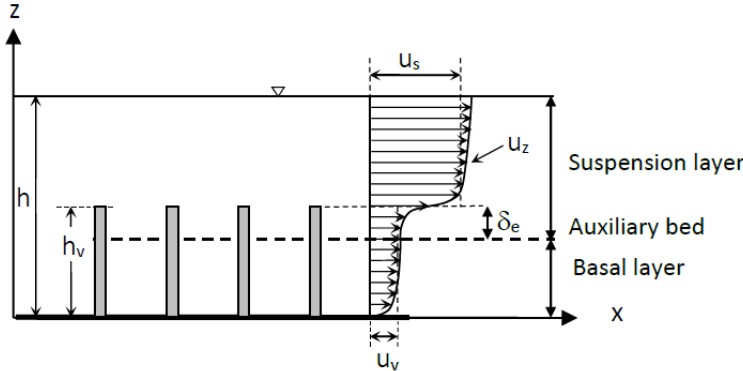

**Figure 6.** Side view of submerged vegetation with auxiliary bed and velocity profile.

Cheng [84], with reference to the flow rates, in the case of rigid submerged vegetation, compared his model with that of Stone and Shen [61], Baptist et al. [62], Huthoff et al. [75], and Yang and Choi [76]. It emerged that his model gave the best overall results while the others exhibited different behavior with varying flow rates. In particular, the Stone and Shen [61] model underestimated the flow rates especially for the higher values, while the model by Baptist et al. [62] works well for high flow rates, but yields overprediction for low flow rates, whereas Huthoff et al.'s [75] model gives the best prediction for high flow rates, but underestimates low flow rates.

Based on the Vargas-Luna et al. [79] analysis, with reference to the comparison between the measured Chézy coefficient and the one estimated with different models [61,62,73,75,76,84,85], it appears that the Klopstra et al. [73], van Velzen et al. [85], and Cheng [84] models show the best results. Although many models are based on a representation of vegetation with rigid cylinders, according to the authors, they also provide good results in the case of flexible vegetation when using the inflected vegetation height.

Pasquino and Gualtieri [81] compared the predicted and measured velocity for the Stone and Shen [61], Huthoff et al. [75], Yang and Choi [76], Cheng [84], Baptist et al. [62], and Li et al. [77] models as the density (sparse vegetation, transitional density, dense vegetation) and the degree of submergence (low submergence and high submergence) varied. They analyzed the performances of the different models on the basis of six different statistical parameters. In the case of transitional density with low submergence, all models give good results; in the case of dense vegetation with low submergence, the Huthoff et al. [75], Cheng [84], Baptist et al. [62], and Li et al. [77] models give very good results. In other cases, the number of data for estimating the statistical parameters, even if this analysis has been carried out, is less than 10.

Morri et al. [80], with the comparison between the measured and simulated mean velocities by different analytical models, highlight how the model by Huthoff et al. [75] provides the best results. It can be used for velocity lower than 0.8 m/s and underestimates the velocity values in the case of sparse vegetation, which could be explained because the bed roughness effect is neglected.

To conclude the analysis of the models proposed for submerged rigid vegetation, it must be said that they refer essentially to laboratory data and give overall good results, even if the models by Cheng [84] and Huthoff et al. [75] seem to provide the best results.

One must note that comparison among the equations presented by various authors is more difficult than in the case of rigid emergent vegetation, because of the many parameters or variables involved.

### 4.1.3. Submerged and Emergent Rigid Vegetation

Katul et al. [87] and Luhar and Nepf [11] proposed expressions to calculate the Manning coefficient for both submerged and emergent vegetation, given in Table 3.

**Table 3.** Equations for estimating *n* in case of submerged and emergent rigid vegetation.

| Authors | Relationship |
|---|---|
| Katul et al. [87,88] | $$n = \frac{h^{1/6}}{\sqrt{g}\,C_u f_c(h/h_v, \alpha_{KA})}$$ with $f_c(h/h_v, \alpha_{KA}) = 1 + \alpha_{KA}\frac{h_v}{h}\ln\left(\frac{\cosh\left(\frac{1}{\alpha_{KA}} - \frac{1}{\alpha_{KA}}\frac{h}{h_v}\right)}{\cosh\left(\frac{1}{\alpha_{KA}}\right)}\right)$ |
| Luhar and Nepf [11] | $n = \frac{k_{LN}h^{1/6}}{g^{1/2}}\left(\frac{C_f}{2}\right)^{1/2}(1-B_x)^{-3/2}$ for $B_x < 0.8$ |
| Luhar and Nepf [11] | $n = \frac{k_{LN}h^{1/6}}{g^{1/2}}\left(\frac{C_D ah}{2}\right)^{1/2}$ for $B_x = 1.0$ |
| Luhar and Nepf [11] | $$n = \frac{k_{LN}h^{1/6}}{g^{1/2}}\frac{1}{\left(\frac{2}{C_f}\right)^{1/2}\left(1 - \frac{h_v}{h}\right)^{3/2} + \left(\frac{2}{C_D ah_v}\right)^{1/2}\left(\frac{h_v}{h}\right)}$$ when the vegetation, of height $h_v < h$, fills the entire width. |

Katul et al. [87,88] starting from the characteristics of the velocity in turbulent flow within and above rigid vegetation canopies proposed to calculate the Manning coefficient according to the relationship reported in Table 3, where $C_u$ is the similarity constant (empirically estimated as 4.5) and $\alpha_{KA}$ is the characteristic eddy size coefficient (estimated as 1 for gravel bed streams and 0.5 for canopy). The previous relationship can also be used in the case of shallow streams; more generally, it is valid for $0.2 < h/D < 7$.

Luhar and Nepf [11], in agreement with Green [10], consider that the flow resistance due to aquatic vegetation depends on the blockage factor, $B_x$, which is the fraction of the channel cross-section blocked by vegetation. Additionally, for the same blockage factor $B_x$, the distribution of vegetation, in terms of distinct patches, affects the hydraulic resistance, since the interfacial area between the patches and the unobstructed flow increases when the patches are divided into smaller units. However, on the basis of observations made in natural rivers, the authors estimate velocities for the case where the blockage factor is known, but the exact distribution pattern is unknown, and it introduces up to 20% uncertainty. Luhar and Nepf [11] propose the relationships reported in Table 3 for $B_x < 0.8$, for $B_x = 1.0$ and for channels where the vegetation, of height $h_v < h$, fills the entire width. In the reported equations, $k_{LN} = 1\ m^{1/3}/s$ is a constant necessary to make the equation dimensionally correct, $a$ is the frontal area per unit volume, $C_f$ is a non-dimensional coefficient proposed by Luhar-Nepf [11] given by $C_f = 2g/C^2$ where $C$ is the Chezy coefficient.

### 4.2 Flexible Vegetation

### 4.2.1. Potentially Changing Vegetation Condition

Sometimes during a flood event, as the discharge increases, the vegetation can lay over or be removed [89–92], which leads to a reduction in roughness and to an increase in the flow capacity through the section; therefore, the peak flow, which could occur later, takes lower water-surface elevations than it would have had in the case of upright vegetation. To determine under what conditions the vegetation flattens, Phillips et al. [89] referred to the stream power, defined as $SP = gRSV$, and to the resistance of the vegetation

characterized by an index defined as the susceptibility index of the vegetation. This index is given by the product of the vegetation flexibility factor, the vegetation blocking coefficient, the vegetation distribution coefficient and, finally, the flow depth coefficient. The vegetation flexibility factor is characteristic of the type of vegetation and is equal to the bending moment determined by computing the product of the moment arm (distance from the bed to the point of application) and the force required to bend the vegetation to 45 degrees from the vertical. Phillips et al. [89], with reference to four different types of vegetation, by means of regression techniques, proposed the equations relating bending moment to vegetation height. The vegetation blocking coefficient takes into account the combined resistance associated with the vegetation and was determined by assigning a weighted value to the estimated percentage of the cross-sectional area of flow that was blocked by vegetation for pre-flow conditions. The vegetation distribution coefficient takes into account the fact that vegetation aligned to the flow direction determines a velocities redistribution and the vegetation is affected by a smaller action, whereas in the case of uniform distribution every element is affected by the same velocity. Finally, the action of the flow to bend the vegetation depends on the water depth in relation to vegetation height. Five different categories have been identified, and the relative coefficients have been assigned according to the ratio between hydraulic radius (approximated with mean flow depth) and vegetation height. Phillips et al. [89], based on the experimental values, identified a threshold curve from which, for an assigned value of the vegetation susceptibility index, it is possible to determine the minimum value of the stream power beyond which the vegetation is layover.

Francalanci et al. [92], with reference to a river in Italy carried out the hydraulic modelling of one flood event in order to investigate the hydraulic forcing on trees, in terms of flow velocity and water depth. A conceptual model for the stability of riparian vegetation, based on the flow drag, the plant weight, and the resistance force of the root apparatus at the soil-root interface, has been used in a predictive form to investigate the range of the input variables which can promote the plant uprooting or preserve its stability.

For rigid vegetation, the action of the flow varies with the squared velocity. In case of flexible vegetation, this is not true, since the vegetation reconfigures by reducing the area projected on a plane orthogonal to the flow direction and aligning the leaves with it. The relationship between flow velocity and drag force was expressed as $F_D \propto V^{2+b}$, where the Vogel exponent $b$ [58] is a measure of the plant reconfiguration. When is $b = -1$, the drag force varies linearly with the velocity. A linear increase of drag force with the flow velocity was observed for flexible plants by direct measurement in prototype scale by Armanini et al. [93].

The friction factor due to the vegetation, $f_v$, in a reach of length $L$ is

$$f_v = \frac{8 F_D h}{\rho V^2 A L} \tag{11}$$

We will first analyze the studies in the case of submerged flexible vegetation and then those related to the non-submerged vegetation.

### 4.2.2. Submerged Flexible Vegetation

The first studies on submerged flexible vegetation (Figure 7) are related to the design of irrigation canals [94]. In Figure 7, $h_{vf}$ is the bent vegetation height. Since the resistance depends on the curvature of the vegetation, the link between the Manning coefficient, $n$, and the product of velocity, $V$, and hydraulic radius, $R$, was identified on an experimental basis. This relationship depends on the type of vegetation and is practically independent on the slope of the canal and its shape. Five experimental curves have been obtained relating the Manning coefficients, also called delay coefficients, with the product VR classified as very high, high, moderate, low, or very low, and identified with the letters A, B, C, D, and E. This link was later reviewed and generalized [95–98].

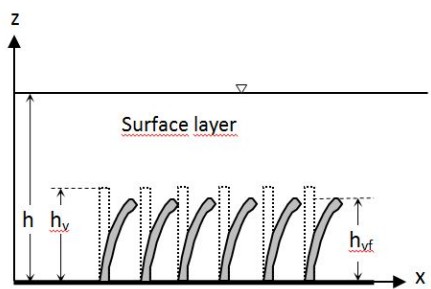

**Figure 7.** Side view of submerged flexible vegetation.

The aforementioned curves, for $n < 0.2$, were interpolated with the following equation:

$$n = \frac{1}{(2.08 + 2.3x + 6\,ln(10.8\,VR))} \tag{12}$$

with $x$ equal to −0.5, 2, 5, 7, 11 respectively for curves A, B, C, D, and E [97].

According to Kouwen et al. [99], the friction factor in the case of submerged flexible vegetation can be represented through a semi-logarithmic function of the relative roughness, defined as the ratio of the deflected plant height, $h_{vf}$, to the flow depth

$$\frac{V}{u_*} = C_0 + C_1 \log\frac{h}{h_{vf}} \tag{13}$$

where the coefficients $C_0$ and $C_1$ depend on $u_*/u_{*c}$ with $u_{*c}$

$$u_{*c} = min(0.23\,MEI^{0.106}, 0.028 + 6.33MEI^2) \tag{14}$$

where $M$ is the stem count per unit area, $E$ is the modulus of elasticity, and $I$ the second moment of the cross-sectional area of the stems. The product $MEI$ is defined as flexural rigidity. The deflected height depends on the drag exerted by the flow and the flexural rigidity of the vegetation. Kouwen and Unny [90] deduced the following relationship:

$$\frac{h_{vf}}{h_v} = 0.14\left[\frac{\left(\frac{MEI}{\gamma hS}\right)^{1/4}}{h_v}\right]^{1.59} \tag{15}$$

where $\gamma$ is the water specific weight.

Kouwen's [100] model implicitly considers the fact that the flow can cause the vegetation to lay over on the bottom of the channel without using the stream power illustrated in point 4.2.1.

Carollo et al. [91], based on the dimensional analysis, wrote the equation

$$\frac{V}{u_*} = \sqrt{\frac{8}{f_v}} = f_c(m)\left(\frac{h}{h_{vf}}\right)^{\beta_1}\left(\frac{u_*h_{vf}}{\upsilon}\right)^{\beta_2}\left(\frac{h_v}{h_{vf}}\right)^{\beta_3} \tag{16}$$

where $f_c$ is a function of the concentration $m$ (number of stems per unit area), $h_v$ the vegetation height in the absence of flow, $h_{vf}$ the height of bent vegetation, and $u_*$ the shear velocity. The authors carried out experimental test to determine the exponents $\beta_1, \beta_2, \beta_3$ and the function $f_c$, which allow computation of the velocity $V$, since the bent vegetation height can be evaluated on the basis of an empirical equation.

Stephan et al. [101] analyzed, through laboratory experiments, the influence of roughness caused by aquatic vegetation on the overall flow field. The analysis of the velocity measurement shows that equivalent sand roughness and zero plane displacement

of the logarithmic law are of the same order of magnitude as the mean deflected plant height.

### 4.2.3. Non-Submerged Flexible Vegetation

Studies on non-submerged flexible vegetation have been carried out, among others, by Kouwen and Fathi Moghadam [102], Freeman et al. [103], Västilä et al. [104], and Jalonen and Järvelä [105]. In particular, Kouwen and Fathi Moghadam [102], Västilä et al. [104], and Jalonen and Järvelä [105] provide an expression for the friction factor, while Freeman et al. [103] provide an expression for the Manning coefficient, given in Table 4.

**Table 4.** Equations for estimating f$_v$ and *n* in case of non-submerged flexible vegetation.

| Authors | Relationship |
|---|---|
| Kouwen and Fathi Moghadam [102] | $f_v = 4.06 \left( \dfrac{V}{\sqrt{\xi E / \rho}} \right)^{-0.46} \left( \dfrac{h}{h_v} \right)$ |
| Västilä et al. [104] | $f_v = 4 C_{D\chi} \dfrac{A_c}{A_b} \left( \dfrac{V}{V_\chi} \right)^{\chi}$ |
| Jalonen and Järvelä [105] | $f_v = \dfrac{4}{A_b} \left[ A_L C_{D\chi,F} \left( \dfrac{V}{V_{\chi,F}} \right)^{\chi_F} + A_S C_{D\chi,S} \left( \dfrac{V}{V_{\chi,S}} \right)^{\chi_S} \right]$ |
| Freeman et al. [103] | $n = 3.487 * 10^{-5} \left( \dfrac{E A_s}{\rho A_i^* \, u_*^2} \right)^{0.150} (m A_i^*)^{0.166} \left( \dfrac{u_* R}{v} \right)^{0.622} \left( \dfrac{R^{2/3} J}{u_*} \right)$ |

Kouwen and Fathi Moghadam [102] performed laboratory experiments on four different types of conifers both in water and in air; the authors, on the basis of dimensional analysis and a series of simplifying hypotheses, proposed to calculate the friction factor as reported in Table 4, where $\xi$ takes into account the deformation of the plant, and $E$ is the modulus of elasticity. The term $\xi E$ is called vegetation index, it is unique for each species and is obtained from the resonant frequency, mass, and length of a tree [102,106]. The authors then propose to suitably modify the resistance to flow when the projection of a plant area overlaps with that of adjacent plants.

Recently, Västilä et al. [104] proposed to calculate the friction factor, $f_v$, as shown in Table 4, where $A_c$ represents a characteristic area of plants, $A_b$ is the bed area related to a plant, $C_{D\chi}$ is a species-specific drag coefficient, $\chi$ depends on the area $A_c$, and $V_\chi$ is the lowest velocity used in determining $\chi$. As to the area $A_c$, Västilä et al. [104] compared three areas: The first one, $A_L$, is obtained by means of the leaf area index (LAI), (defined as the ratio between the one-sided leaf area and the ground area), the second one as the area projected onto a plan orthogonal to the flow direction when the plant is under the flow action ($A_p$), and the third one as the area projected onto a plan orthogonal to the flow direction when the plant is on free air ($A_0$). The results obtained by Västilä et al. [104] show that the three above characteristic areas give good results, even though the easiest area to quantify is the one relative to the leaf area index, that can be measured by laser-scanner techniques. Västilä et al. [104] determined the parameter values for Black Poplars ($C_{D\chi} = 0.33$, $V_\chi = 0.1$, $\chi = -1.03$, $0.4 < A_L/A_b < 3.21$) and other foliated deciduous species of Poplars ($0.43 \leq C_{D\chi} \leq 0.53$, $0.10 \leq V_\chi \leq 0.11$, $-0.57 \leq \chi \leq -0.9$, $0.4 \leq A_L/A_b \leq 3.2$).

Jalonen and Järvelä [105], taking into account the reconfiguration properties of branches and leaves, split their contribution in the friction factor proposing the equation reported in Table 4, where the subscript $F$ and $S$ refer to leaves and stem, respectively. The authors give the values of the parameters ($C_{D\chi,F}, \chi_F, C_{D\chi,S}, \chi_S$) for four different species, in the velocity field from 0.1 m/s to 0.6 m/s (low) and for velocities higher than 0.6 m/s (high).

Freeman et al. [103] on the basis of dimensional analysis and extensive experimental investigation, wrote the Manning coefficient as reported in Table 4, where $A_i^*$ is the net submerged frontal area, $A_s$ is the total cross-sectional area of all of the stems of an indi-

vidual plant, measured at a quarter of its non-deflected height, m is the density. The plant characteristic $A_i^*$ and $A_s$ are density-weighted average values that were measured without any bending or distortion attributable to flow. The modulus of elasticity is calculated starting from the force necessary to bend the plants at an angle of 45° or, more simply, by an empirical relation expressed as a function of the ratio between the height of the plant and its diameter measured at a quarter of the height.

Once again, for submerged or non-submerged flexible vegetation, comparison among the equations proposed by different authors is difficult if not impossible.

## 5. Numerical Methods

Apart the empirical approach, turbulence study is the basis for flow resistance assessment, at least from Prandtl and von Kàrmàn theory dating back to the beginning of the last century. Moreover, a detailed analysis of the flow fields and turbulence characteristics may be of importance as related to solid and pollutant transport.

Experimental studies of velocity profiles, like those shown in Figures 3–5, are of much interest; in some cases also the shear stress profiles are studied by means of velocity fluctuant components and Reynolds stress theory [19,107–111].

For a detailed analysis of the flow fields and turbulence characteristics, one can refer to measurements with a particle image velocimetry system (PIV) or acoustic Doppler velocimeter (ADV) probe or, furthermore, on numerical simulations, more or less detailed depending on the flow cases at hand [65,112–115]. An excellent review about the numerical models utilized for the analysis of the interaction between flow and vegetation is due to Stoesser et al. [116], and one can refer to this work for a systematic view of the different aspects of this matter. In what follows, attention is given to the more recent works on this subject.

The interaction between flow and vegetation has been studied by means of RANS (Reynolds Averaged Navier-Stokes equations [112]) and LES (Large Eddy Simulation [117]) techniques. The DNS (Direct Numerical Simulation of turbulence [118]) approach was used to analyze the interaction between fluid and cylinders, in which all the fluctuating components are computed and no closure models are used, then the swirling-strength criterion for flow-structure extraction is applied to the velocity field [119]; this method is scarcely used, mainly due to the remarkable computing resources required.

RANS models are operated on coarse grids, and the drag due to vegetation is accounted for through additional source terms in the governing equations [86,120–123]. These models require an a-priori knowledge of the drag coefficient, contain a number of empirical constants, and are fairly accurate as related to the evaluation of the mean flow field, but do not simulate the flow past single trunks or branches. Kim and Stoesser [124] demonstrated the importance of the a-priori knowledge of the drag coefficient for the correct evaluation of the resistance to flow. Their RANS simulations, carried out by adopting actual $C_D$ values, furnished values of the tangential shear stress on the bottom in a perfect agreement with the experimental results, while the adoption of a constant unitary value produced an underestimation of the resistance to flow as the density was greater. The coherent structures that form downstream to the cylinders or over the vegetated layer in the case of submerged vegetation can be obtained by means of the LES approach [72,117,124]. In the latter simulations, one can still refer to the resistance to flow of each single plant through the drag coefficient, or one can explicitly represent the plants by inserting in the mesh some blocked cells representing the vegetation. Stoesser et al. [72] executed a LES with reference to a companion experiment of Liu et al. [59] related to rigid emergent vegetation, obtaining interesting results. They calculated the flow characteristics with reference to reach 0.127 m long and 0.0635 m wide, with 11,604,640 computational cells. The time-averaged velocity profiles along six vertical lines resulted in a satisfactory agreement with the experimental values, both as to the streamwise and the normal components. The velocity field showed a high-velocity zone between the

cylinders, the flow separation at 95°, and a recirculation zone downstream to the cylinders with two counter-rotating vortices. Additionally, the turbulence intensities in the streamwise and vertical directions showed a good agreement with the experimental results. Then the authors executed additional simulations with reference to densities and Reynolds numbers different from the experimental case. Starting from mean velocities and pressure fields, Stoesser et al. [72] calculated the drag on the cylinders, as decomposed in pressure and shear, and the shear on the bed. The drag coefficient values resulted in a good agreement with the experimental values of Tanino and Nepf [66]. Turbulence was investigated through the instantaneous pressure fluctuations. Kim and Stoesser [124] proposed a low-resolution LES calculation in which the computing time is considerably reduced compared to a fully resolved LES. In a high-resolution LES, each vegetation element is represented by means of a curvilinear grid, while in the case of low-resolution LES, the vegetation is handled by means of a simplified immersed boundary condition on a Cartesian grid (see Figure 2 of Kim and Stoesser [124]). A square-cell grid is preliminarily built-up. In the area interested by a generic circular obstacle, one can distinguish three types of cells, *i)* those belonging to the obstacle for which the velocity is zero, *ii)* those belonging to the fluid, for which the velocity is computed without any treatment, and *iii)* those belonging to both of them (cut cells), for which the computed velocity is multiplied by the "volume fraction" (the volume of fluid divided by the total cell volume). Kim and Stoesser [124] compared the results of the low-resolution calculations with those of high-resolution calculations (different densities and Reynolds numbers of 500 and 1340 have been considered as related to a single simulation, to 534,681 and 22,994,560 grid points, respectively) and found that, in particular for low densities, the model was able to correctly represent the velocity gradients, the wakes behind the stems, and the secondary flows. In the low-resolution LES, the knowledge of the drag coefficient was not necessary, and the model could be used also to analyze complex rod configurations, namely casual distribution of the vegetation with elements very near to each other.

It should be noted that numerical simulations have been used not only for comparisons with companion experiments, but also in real situations. As an example [116,125], a RANS simulation was carried out to mirror a flood event of the Rhine river with a return period of 100 years. A reach of 3.46 km was considered, as discretized with a mesh of 258 × 64 × 12 computational cells along the streamwise, spanwise, and vertical directions, respectively, for a total of 198,144 cells. The dimension of each cell was about 13 m × 3 m × 0.5 m. The resistance to flow due to the plants was taken into account only in the momentum equation, the characteristic of the vegetation was taken into account starting from a survey related to an area of 77 m × 45 m, the Manning coefficient was calibrated on the basis of a flood event that not involved the floodplain, while the representative diameter of the vegetation was calibrated on the basis of a flood event that actually flooded the whole floodplain. The flow velocity was measured using dilution gauging techniques. The numerical results were in good agreement with the observed levels, and the values of the velocities in the zone with vegetation corresponded well to those measured by the dilution method. Additionally, the velocity distribution in the whole cross section resulted in a good agreement with the expected values. In conclusion, the turbulence study based on numerical approach is promising, even though still very far from practical use.

## 6. Hydraulic Roughness Assessment

Riparian vegetation, in particular that located on flood areas, has very heterogeneous characteristics, both from a spatial and temporal point of view. These characteristics must be adequately included in the hydraulic–hydrological models. Conventional ground-based monitoring is often unfeasible, as these techniques are time-demanding and expensive [126], especially for large areas and when they are inaccessible. New opportunities are offered by remote sensing, which has developed considerably in recent

decades and has been increasingly used in the environmental field. Some reviews have addressed the use of remote sensing in fluvial studies [127–129] and, in particular, for mapping riparian vegetation and estimating biomechanical parameters [130].

Remote sensing is based on satellite images (digital or radar) or aerial platforms (LiDAR (Light Detection and Ranging) and orto-photography). Digital satellite images, in the last decade, have reached a definition similar to those of orthophotos (the pixel sizes in the sensors Quickbird and Ikonos are equal, respectively, to 0.6 m and 1 m) making possible their application to riparian areas that, very often, are of limited size. Image classification is the process of assigning individual pixel or groups of pixels to thematic classes; it is either supervised and unsupervised.

The classification of the images is very often supervised, that is, it is based on a priori knowledge of the type of coverage, but in the case of very high resolutions, it is based on segmentation techniques [131]: Edge-finding, region-growing, knowledge-based segmentation, and a mixture of the last two.

Forzieri et al. [132] proposed a method to estimate the vegetation height and flexural rigidity for the herbaceous patterns and plant density, tree height, stem diameter, crown base height, and crown diameter of high-forest and coppice consociations for arboreal and shrub patterns from satellite multispectral data (SPOT 5). The method is developed through four sequential steps: (1) Classification of pixel surface reflectance into five land cover classes: Mixed arboreal, shrub, herbaceous, bare soil, and water; (2) data transformation based on Principal Component Analysis of the original multispectral bands and use of only the first principal component since it explains a lot of variances; (3) identification of significant correlation structures between the main components and biomechanical properties; (4) identification/estimation/validation of the relationship (simple tri-parametric power laws) between the biomechanical properties and the normalized principal component. The vegetation hydrodynamic maps are also able to well describe the equivalent Manning's roughness coefficient as proved by comparison with simulated water stages [132].

One of the biggest limitations of optical sensors is the inability to penetrate the cloud system. Radar systems are microwave-based and do not depend on the cloud system, and are particularly useful during flooding events that usually occur in the presence of a cloud cover, allowing monitoring of the timing and spatial extent of flooding. Backscatter increases with biomass, and this makes it difficult to apply radar sensors in floodplain areas, which are usually characterized by very dense vegetation.

Satellite images provide information on the spatial variability of vegetation, but do not provide information about its vertical structure. LiDAR technology provides information on the three-dimensional structure of vegetation. Laser scanning (LS) is employed in terrestrial (TLS), airborne (ALS), and mobile (MLS) platforms. The airborne laser scanner (ALS) provides accurate information of forest canopy and ground elevations producing a digital terrain model and a digital surface model. The difference between the digital surface model and the digital terrain model gives the tree heights. Forzieri et al. [133] developed a model to identify individual tree positions, crown boundaries, and plant density using airborne LiDAR data. It needs an initial calibration phase based on a multiple attribute decision making simple additive weighting method. Jalonen et al. [134] employed multi-station TLS, both in field and laboratory conditions, to derive the total plant areas of herbaceous vegetation and the vertical distribution of the total plant area of foliated woody vegetation for different levels of submergence.

## 7. Flow Resistance and Vegetation Management

The different approaches and methods described above can be used in 1D, 2D, and 3D models.

These models can be useful in a multidisciplinary framework that includes hydrology, hydraulics, forestry, sediment transport, and ecology, which are the best vegetation management strategies. It is practically impossible to analyze the latter in the field, since

one should see the flow behavior for the design discharge. To our knowledge, the only field experiments of different vegetation management is that of Errico et al. [135], who compared the effects of three different vegetation scenarios on flow velocity distribution, turbulence patterns, and flow resistance in drainage channel colonized by common reeds. The first scenario corresponded to the canopy in undisturbed conditions, the second and the third scenarios were obtained by clearing, respectively, the central part and the entire channel. The channel conveyance was obtained by clearing reeds in just the central part of the drainage channel and was comparable to that obtained by the total clearance, but with much less ecological impact and maintaining relatively high levels of turbulent intensities. Errico et al. [135] also evidenced that the most suitable models for representing natural reed canopies are those that quantify the blockage factor of the patch, rather than the effect of plant elements matrixes.

Phillips and Tadayon [90] were among the first to present some examples of vegetation management in both artificial and natural trapezoidal-shaped channels. Simulation results using HEC-RAS indicate that the design discharge for the channel for full-grown vegetation conditions would overtop channel banks and flood adjacent areas. Instead, simulations conducted for post-vegetation maintenance conditions, consisting in its partial removal, indicate that the design discharge would remain within the channel. A common element to some of the examples shown was verifying whether the design flow has the power to lay over the bushes. If this occurred, the associated roughness component was considered negligible and not used in the determination of the Manning coefficient. They also considered a minimum amount of 30 cm of freeboard above the design water-surface elevation. The purpose of this freeboard is to mitigate risk by providing a factor of safety. As part of the various vegetation maintenance schemes or scenarios, Phillips and Tadayon [90] have highlighted how leaving the vegetation randomly distributed is aesthetically more pleasant, while leaving the trees and bushes grouped together may present a better habitat environment for wildlife.

D'Ippolito and Veltri [136] analyzed the influence exerted by vegetation on the discharge with a return time of 200 years in the mountain reach of the Crati river (Calabria, Italy). The flow profiles, determined by the use of HEC-RAS, showed that, in the reach, there is an average increase in water heights of 5% with maximum peaks of 24%. The influence exerted by herbaceous vegetation is negligible compared to that due to woody vegetation.

Luhar and Nepf [11] assert that mowing patterns that produce less interfacial area per channel length (e.g., a single continuous cut on one side of the channel) are the most effective in reducing hydraulic resistance.

Van der Sande et al. [131] created a detailed land cover map of the villages of Itteren and Borgharen in the southern part of the Netherlands by using IKONOS-2 high spatial resolution satellite imagery and combining the use of spectral, spatial, and contextual information. It is one of the first applications of land cover maps used as inputs for flood simulation models. The plant cover is classified in three classes: Natural vegetation ($n = 0.1$), deciduous forest ($n = 0.2$), and mixed forest ($n = 0.2$). Using the above maps, the model results are not very different from those obtained using less accurate maps. Van der Sande et al. [131] conclude that their use provides better results (flow direction and water depth) for less extreme events.

Abu Aly et al. [88], with reference to a gravel-cobble river from California, analyzed the effects of vegetation on velocities, depths, and extent of the flooded areas for flows ranging from 0.2 to 20 times the bankfull discharge ($Q_{BF}$). They used a two-dimensional finite volume model that solves the vertical-mediated Reynolds equations (SRH-2D) and analyzed a stretch of the water course 28.3 km long with a mesh of 1–3 m. They obtained the height of the vegetation in each cell from LiDAR data and estimated the Manning coefficient based on the approach developed by Katul et al. [87] reported above. Compared to the case of absence of vegetation, they achieved an increase in the mean water depth of 7.4% and a mean velocity decrease of 17.5% for a flow of 4 $Q_{BF}$, values that rise,

respectively, to 25% and 30% for a flow rate of 22 $Q_{BF}$. The model also shows how the vegetation has a strong channelization effect on the flow, in fact the flow is diverted away from densely vegetated areas and there is an increase in the difference between mid-channel and bank velocities.

Benifei et al. [137] investigate the effects of channel and riparian vegetation on flood event, with 200-year-return-period, really happened. Since the water levels were strongly influenced by riparian vegetation on the floodplain, they analyzed the effectiveness of different flood mitigation strategies employing the model Delft3D. They use the Baptist method [62]. Based on the results of the model, Benifei et al. [137] show that the most effective flood management strategy is obtained when the high vegetation (composed by trees) inside the flooding area with a return period of 2 years is removed, avoiding the growth of the bushes (5-years-old trees) as well.

## 8. Future Research and Conclusions

The paper is a review of a considerable number of studies on the flow resistance due to vegetation, based on experimental investigations in the presence of simulated or real vegetation; in addition, the paragraph on numerical methods also refers to phenomena on the local scale. It should be said, however, that flow resistance is a matter of detail because, according to Tsujimoto [138], flow, sediment transport, geomorphology, and vegetation constitute an interrelated system. The analysis of this interaction requires a multidisciplinary approach that can involve a set of disciplines such as hydraulics, hydrology, sediment transport, ecology, botany, and geotechnical engineering. It should also be noted that these phenomena vary greatly over time, in fact the climate and hydrology can alter growth patterns, rates of colonization, and vegetation density. The vegetation is simultaneously a dependent variable and an independent variable. Recent research has begun to investigate the interaction between sediment and vegetation in terms of sediment transport, erosion, and deposition, as stated in the introduction, but these interactions remain poorly quantified [16] and are new broad fields of research; the same is to be said for the interaction between vegetation and river morphodynamics [139,140].

With reference to flow resistance in the case of rigid vegetation many arrangements have been investigated taking into account different variables, so that a comparison of the proposed formulas is not immediate, even though possible. The first studies started from the use of basic equations, like momentum equation, and concepts, like the drag coefficient. Later, the studies developed towards the velocity and shear stress distribution. In the case of rigid vegetation, a few variables look to be able to derive a law for flow resistance. In the case of flexible vegetation, on the other hand, the various investigations refer to specific shrub and tree varieties, and the parameters of the different formulas are related to the specific varieties investigated, as well to the measuring range of the experimental tests. The wide variety of vegetation types and hydrodynamic conditions make comparing results and drawing general conclusions that are useful in practice difficult. Shields et al. [141] present a comparison between the Manning coefficients calculated, among others, with some of the formulas presented in the previous paragraphs both in the case of rigid vegetation and flexible vegetation. From it emerges how the formulas considered *predict values within the correct order of magnitude, however, clear guidelines for selecting one approach over another are difficult to provide* [141]. These indications could be obtained, as will be better specified later, seeing the ability of the different models to reproduce real situations. For operational purposes and for the correct use of formulas, it would be useful to establish a reference catalogue in which vegetation should be classified as rigid or flexible and in which ranges of biomechanical properties (flexural rigidity): Moreover, the geometric characteristics (shape of plants, foliage density, leaf area index) of the different species present in the river bed, on the banks, and in the floodplain areas, should be reported even better identifying how these properties vary with an appropriate parameter (age, height, or other). The vegetation, both in terms of

species and density, varies considerably in space and in time. The hydraulic models that can best take into account this variability are the two-dimensional models. Remote sensing can be used to define the different roughness values, which is particularly useful in large areas and/or in areas not easily reachable. To accurately estimate the main vegetation properties, the simultaneous use of different remote sensing techniques could be used. Although methodologies have been proposed to determine height, flexural rigidity, density, stem diameter, crown base height, and diameter of vegetation, this constitutes a field with broad development perspectives. Very often it is difficult to classify the different types of vegetation, and remote sensing data require calibration through direct analysis in the field.

In order to identify which of the models previously seen are the most suitable to be used in the verification and design of interventions, it would be appropriate to apply them to real cases to reproduce flood events; an example of this is that of Benifei et al. [137], even if it is limited to the use of only one model. Considering the danger of flood events, reference could be made to controlled floods or indirect measures. At the same time, the same models can be used for the field verification of different riparian vegetation management strategies [135] and, subsequently, after calibration, for the design of the interventions.

Studies should be carried out on the dynamics of flood phenomena in the sense that high discharge can cause vegetation to bend on the bottom or break it or, due to the effect of localized erosion, can determine the uprooting of plants. Consequently, peak discharge or receding flow can have lower heights, even though this is valid only from a theoretical point of view because the material transported can accumulate on the bridge piles, causing dangerous backwater effects.

While the evaluation of flow resistance with reference to river reach is of considerable importance, it must be said that even the effect of vegetation patches exert considerable influence on flow resistance, on velocity distribution, and solid transport, and this is also a field of further research as witnessed by recent publications [34–38]. The kinetics of vegetation in terms of establishment, growth, and decay of riparian vegetation in response to dynamic hydraulic conditions are also beginning to be of interest to researchers [45,142].

While the present available results are mainly of global type, detailed numerical simulations can give insights on the characteristics of the flow fields when they interact with vegetation. This is also not a simple issue, due to the different properties that the vegetation can exhibit (rigid, submerged, etc.). Actually, the 3D numerical models can simulate very simple arrangements, and the currently required calculation resources go far beyond those used in professional practice. However, the continuous progresses in the numerical field and software development can guarantee that the detailed calculation of the flow–vegetation interaction will be soon possible. This issue involves both direct numerical simulations and numerical modeling of turbulence in this context, which may also benefit from the relevant theoretical developments and progress in observation techniques. So, in the future, they can provide more useful information for the conservation of habitats and biodiversity, maintenance of vegetation, and flooding protection.

**Author Contributions:** Conceptualization, F.C., A.D.; methodology, F.C., A.D.; writing—original draft preparation, F.C., A.D.; writing—review and editing, F.C., G.A., A.D., A.L.; supervision, F.C. All authors have read and agreed to the published version of the manuscript.

**Funding:** This research received no external funding.

**Conflicts of Interest:** The authors declare no conflict of interest.

**Notations**

The following symbols are used in this paper:

| | |
|---|---|
| $A$ | area of water cross-section |
| $a$ | projected plant area per unit volume |
| $A_0$ | plant area projected onto a plan orthogonal to the flow direction when the plant is on free air |
| $A_b$ | bed area related to a plant |
| $A_c$ | characteristic area of plants |
| $A_i$ | projected area of the ith plant on a plane normal to the streamwise direction |
| $A_i^*$ | net submerged frontal area of the plant in the plane normal to the flow direction |
| $A_L$ | one-sided leaf area |
| $A_P$ | plant area projected onto a plan orthogonal to the flow direction when the plant is under the flow action |
| $A_S$ | total cross-sectional area of all of the stems of an individual plant, measured at a quarter of the non-deflected height of the plant |
| $b$ | Vogel exponent |
| $B_x$ | fraction of channel cross section blocked by vegetation |
| $C$ | Chézy coefficient |
| $C_b$ | Chézy coefficient of the bed |
| $C_D$ | drag coefficient |
| $C_{D\chi}$ | species-specific drag coefficient |
| $C_{D\chi,F}$ | species-specific drag coefficient of leaves |
| $C_{D\chi,S}$ | species-specific drag coefficient of stems |
| $C_f$ | Luhar-Nepf [11] friction coefficient |
| $C_k$ | Chézy coefficient in the presence of emergent vegetation |
| $C_r$ | Chézy coefficient in the presence of submerged rigid vegetation |
| $C_u$ | coefficient in the Yang and Choi [76] model and in the Katul [87] model |
| $C_0, C_1$ | coefficient in the Kouwen model |
| $D$ | vegetation diameter |
| $E$ | modulus of elasticity |
| $f$ | Darcy Weisbach friction factor |
| $f_c$ | function |
| $F_D$ | drag force |
| $f_v$ | friction factor due to vegetation |
| $g$ | gravity acceleration |
| $h$ | water depth or depth of the immersed part of the cylinder |
| $h_v$ | vegetation height or vegetation height in the absence of flow |
| $h_{vf}$ | bent vegetation height |
| $h_*$ | representative length in Li et al. model [77] |
| $i$ | bed slope |
| $I$ | second moment of the cross-sectional area of the stems |
| $J$ | energy line slope |
| $K$ | Gauckler-Strickler velocity coefficient |
| $L$ | length of river reach |
| $LAI$ | Leaf area index |
| $M$ | number of stems per unit bed area |
| $m$ | number of cylinders per unit bed area |
| $MEI$ | flexural rigidity |
| $n$ | Manning roughness coefficient |
| $n_b$ | soil Manning roughness coefficient |
| $Q$ | discharge |
| $R$ | hydraulic radius |
| $Re_D$ | stem Reynolds number ($=VD/\nu$) |
| $Re_{D*}$ | vegetation Reynolds number calculate with the average pore velocity ($=V_v D/\nu$) |
| $Re_v$ | vegetation Reynolds number ($=V_v r_v/\nu$) |
| $r_v$ | vegetation-related hydraulic radius |
| $r_v^*$ | dimensionless vegetation-related hydraulic radius |
| $s$ | separation between individual resistance element |
| $S$ | energy slope |

SP　　　　　stream power
u　　　　　　mean velocity along the vertical
$u_s$　　　　mean velocity along the vertical in the surface layer in the case of submerged vegetation
$u_v$　　　　mean velocity along the vertical in the vegetated layer in the case of submerged vegetation
$u_z$　　　　local time-averaged velocity
$u_*$　　　　shear velocity
$u_{*c}$　　　vegetal critical shear velocity
V　　　　　　mean flow velocity or approach velocity
$V_v$　　　　average pore velocity
$V_\chi$　　　lowest velocity used in determining $\chi$ in Västilä et al. [104] model
x　　　　　　streamwise coordinate
z　　　　　　vertical coordinate
$\alpha_{0E}$, $\alpha_{1E}$ coefficients in the Ergun relationship for the drag coefficient
$\alpha_{KA}$　　characteristic eddy size coefficient in the Katul et al. [87] model
$\alpha_{KL}$　　characteristic turbulence length scale in the Klopstra et al. model [73]
$\beta_1$, $\beta_2$, $\beta_3$ numerical coefficients in the Carollo et al. [91] model
$\chi$　　　　vegetation parameter in Västilä et al. [104] model
$\chi_F$　　　parameter in Västilä et al. [104] model relative to leaves
$\chi_S$　　　parameter in Västilä et al. [104] model relative to stems
$\delta_e$　　　depth of penetration of suspension layer in vegetation in Li et al. [77] model
$\gamma$　　　　water specific weight
$\lambda$　　　　density of vegetation
$\nu$　　　　　water kinematic viscosity
$\xi$　　　　　parameter that takes into account the deformation of the plant
$\xi E$　　　　vegetation index
$\rho$　　　　　water density

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
