# Peer review of "Flow Resistance in Open Channel Due to Vegetation at Reach Scale: A Review"

_water, doi:10.3390/w13020116_

Round 1

Reviewer 1 Report

This paper provides a good review on the hydrodynamic resistance in open channel with vegetation bottom/bank. It almost covers the key and typical references in this field / on this topic. I would suggest to accept after addressing some minor issues as follows:

Line 58: Is ‘in literature’ & ‘in the literature’ repeated?

Line 205-206: Why ‘the diagonal distributions determine a greater resistance than the linear’? Can you replenish a brief explanation of this phenomenon? Or is it a common conclusion? Are there other similar results in previous studies?

Line 225-226: Why ‘impose the equality between the weight of the control volume, projected on the bed direction plus the contour resistance and that opposed by the tree trunks’? What is the point of doing this and what potential influences might this action make?

Line 286: Is ‘others Author’ a misrepresentation? The word ‘author’ is also capitalized elsewhere in the paper.

Line 723-725: What does ‘diverting it away from densely vegetated areas’ mean? What does ‘it’ refer to?

Line 766-767: Why the best hydraulic models are the 2-D models instead of the 3-D ones?

Last but not least, despite the systematical review of classic and typical papers on this topic in the paper, still more recent/latest papers (such as published in recent 2~3 years) are worthwhile to be included. I believe it is not difficult for authors to find out them via online searching or from typical journals such as Environmental Fluid Mechanics, Journal of Hydro-environment Research, Journal Hydraulic Engineering, International Journal of River Basin Management, etc...

Reviewer 2 Report

The manuscript aims to review the relevant literature concerning the evaluation of open-channel flow resistance at the reach scale. The topic is of interest and well suited to the audience of the Journal. The proposed objective is very ambitious since different kinds of vegetation are taken into account (fixed and flexible, submerged and emergent) and various approaches are considered, namely experimental, theoretical and numerical. To fulfill this aim, more than 120 references are described and commented, denoting a rather exhaustive coverage of the relevant bibliography.

The review is formally well-organised, but it lacks a comprehensive effort to achieve a synthesis of the literature reviewed.

Despite few attempts to compare different formulations of the same parameter (see for instance fig. 4), most of the discussion resembles a compendium of results obtained by different Authors under different conditions. Admittedly, in several cases the different approaches are described as non-comparable and the Authors may not to be blamed for this, but as an interested reader the manuscript left me with several open questions:

  • Which are the intrinsic and extrinsic motivations leading to such a variety of results? Could it be expected that these uncertainties will reduce as research advances? Or will they further proliferate?
  • As a practitioner, which approach/method should be used to address a given problem?
  • As a researcher, which are the most promising approaches or ideas to improve the current knowledge in the field?

Ideally, the Authors of a review should be able to provide adequate answers to similar questions. Realistically, and especially for wide and relatively recent research topics (as in the present case), one must accept that this could not be viable, but an attempt to address these questions - even if not completely successful - should be clearly apparent from the review. My impression is that the present manuscript partially misses this prerogative.

My feeling is that the Authors have the required expertise and proficiency to try to address the following issues and to increase, in this way, the potential in the manuscript. I therefore ask them to provide a revision of the manuscript trying to address the following issues.

Additionally, the manuscript deserves further editing, addressing the language (for instance, “emerging” is used often in place of “emergent”, and a moderate revision should be needed to render many sentences in a more concise style) and several minor editorial issues. Many of them are marked in the annotated manuscript attached to the review, which should be considered as a non-exhaustive starting point.

Round 2

Reviewer 1 Report

The authors have revised the paper accordingly, which is now acceptable.

Reviewer 2 Report

The Authors have addressed to an adequate extent the points discussed in the previous revision round. While the main structure of the review has been retained (along with its few limitations addressed in the previous revision), I believe that the manuscript may be accepted for publication.